# Assessing the uncertainty of soil moisture impacts on convective precipitation using a new ensemble approach

Olga Henneberg[1], Felix Ament[2], and Verena Grützun[2]

[1]Niels Bohr Institute, Copenhagen University
[2]Meteorological Institute, University of Hamburg

*Correspondence to:* Olga Henneberg (olga.henneberg@alumni.ethz.ch)

**Abstract.** Soil moisture amount and distribution control evapotranspiration, and thus impact the occurrence of convective precipitation. Many recent model studies demonstrate that changes in initial soil moisture content result in modified convective precipitation. However, to quantify the resulting precipitation changes, the chaotic behavior of the atmospheric system needs to be considered. Slight changes in the simulation setup, such as the chosen model domain, also result in modifications to the simulated precipitation field. This causes an uncertainty due to stochastic variability, which can be large compared to effects caused by soil moisture variations. By shifting the model domain, we estimate the uncertainty of the model results. Our novel uncertainty estimate includes ten simulations with shifted model boundaries and is compared to the effects on precipitation caused by variations in soil moisture amount and local distribution. With this approach, the influence of soil moisture amount and distribution on convective precipitation is quantified. Deviations in simulated precipitation can only be attributed to soil moisture impacts if the systematic effects of soil moisture modifications are larger than the inherent simulation uncertainty at the convection resolving scale.

We performed seven experiments with either modified soil moisture amount or distribution to address the effect of soil moisture on precipitation. Each of the experiments consists of ten ensemble members using the deep convection resolving COSMO model with a grid spacing of 2.8 km. Only in experiments with very strong modification in soil moisture do precipitation changes exceed the model spread in amplitude, location or structure. These changes are caused by a 50% soil moisture increase in either the whole or part of the model domain or by drying the whole model domain. Increasing or decreasing soil moisture both predominantly results in reduced precipitation rates. Replacing the soil moisture with realistic fields from different days has an insignificant influence on precipitation. The findings of this study underlines the need for uncertainty estimates in soil moisture studies based on convection resolving models.

## 1 Introduction

Convective precipitation changes rapidly in space and time (Pedersen et al., 2010). The heterogeneity of convective precipitation and the interaction of different scales is a big challenge in atmospheric models on the global and regional scale. Nowadays, regional climate models operate with a horizontal resolution of 1 km and can represent convective processes explicitly to improve weather forecasting (Mass et al., 2002). Nevertheless, precipitation formation results from a complex chain of at-

mospheric processes, which range from the microscale to the synoptic scale (Richard et al., 2007). Because many of these processes remain unresolved, precipitation is a highly uncertain quantity.

The soil moisture content determines the partitioning of turbulent heat fluxes between sensible and latent heat flux. Depending on land surface properties it controls how much energy is used to heat up the surface or to moisten the atmosphere. The surface temperature plays a crucial role in the initiation of convection, whereas the specific water content in the boundary layer modifies moist conditional instability. On the one hand, low soil moisture content enables fast surface heating, resulting in high surface temperature which can initiate convection. On the other hand, high soil moisture can destabilize the atmosphere by introducing water vapor in the lower troposphere resulting in an enhanced possibility for convection. There is no distinct effect from soil moistening or drying on precipitation intensification, yet there exists a strong systematic influence of soil moisture changes on latent and sensible heat fluxes as well as on equivalent potential temperature, lifting condensation level, and convective energy (Barthlott et al., 2011). Despite these systematic effects, precipitation reacts less systematically to soil moisture variations (Barthlott and Kalthoff, 2011; Hohenegger et al., 2009). The distribution and inhomogeneity of soil moisture patterns may even initiate secondary circulation (Clark et al., 2004; Adler et al., 2011; Kang and Bryan, 2011; Dixon et al., 2013; Maronga and Raasch, 2013; Froidevaux et al., 2014).

There is no clear agreement on the sign of soil moisture - precipitation interaction in the literature. By varying soil moisture by $\pm 25\%$ Barthlott et al. (2011) simulated precipitation changes larger than 500% in regions with low mountain ranges, and changes of up to $-75\%$ for domains with higher mountain ranges. They could not identify significant differences between planetary boundary layer driven and synoptically forced conditions. Hauck et al. (2011) determined large systematic differences between simulated and observed soil moisture. The influence on simulated precipitation in their study was complex and strongly dependent on the particular cases and domains. A dependency of all convective indices on the equivalent potential temperature was found by Kalthoff et al. (2011) over different orography. However, convection was predominantly initiated over mountain crests, independently of the instability indices, but with smaller convective inhibition (CIN). The dependency of equivalent potential temperature on soil moisture was found to be influenced by surface inhomogeneity. Barthlott and Kalthoff (2011) provide a sensitivity study in which the soil moisture was changed by $\pm 50\%$ in steps of 5%. While the study reveals a systematic effect on the 24 hours total precipitation sum for reduced soil moisture, precipitation is not systematically modified by increased soil moisture.

Large variations in these results may partly be attributed to model uncertainty. Hohenegger and Schär (2007) investigated the error growth of random perturbation methods in cloud-resolving models using time shifted model simulations and perturbed temperature fields in the initial conditions. In their model study, using a model resolution of $2.2\,\mathrm{km}$, a rapid error growth was found far away from the perturbed regions, but growth of uncertainties is limited by the large-scale atmospheric environment. A further aspect causing model uncertainty is model resolution, especially regarding the influence on convection. Different results for soil moisture-precipitation feedback also appear in simulations with explicitly resolved and differently parametrized convection (Hohenegger et al., 2009). Hohenegger et al. (2008) found different results in sign and strength of the influence of soil moisture that depend on the model resolution. Simulations with explicitly resolved convection indicate a negative soil moisture-precipitation feedback which is in agreement with many other studies, summarized by Barthlott and Kalthoff (2011).

In numerical weather prediction models, soil moisture perturbations are used to generate ensemble members. Weather services include soil moisture perturbation in data assimilation for their ensemble forecast systems. For example, MeteoSwiss uses the method described by Schraff et al. (2016) in the COSMO model to achieve a model spread, especially in summer (pers. com. Daniel Leuenberger, MeteoSwiss). The evaluation of the AROME-EPS (MeteoFrance) ensemble prediction system presented by Bouttier et al. (2016), which also includes soil moisture perturbations, shows a lack of spread in forecasted precipitation rates. Thus the question is raised as to whether soil moisture perturbations can cause sufficient differences in simulated precipitation. This question will be addressed in the present study. As Richard et al. (2007) stated, convective precipitation output strongly depends on the model setup, such as the prescribed initial conditions and boundary data. In the present study, we provide a description of changes in simulated precipitation resulting from a different amount or a changed pattern of soil moisture together with an assessment of the uncertainty in precipitation caused by random processes in the model. The uncertainty is estimated from an ensemble generated with different boundary conditions by slightly shifting the model domain. Based on a large number of simulations with slightly changed model setup, the systematic influence of different soil moisture modifications on precipitation can be identified and quantified.

Model simulations are conducted with the regional model COSMO (section 2.1). The soil moisture experiments and the ensemble approach are presented in sections 2.2 and 2.3, respectively, for a case study with convective precipitation. An overview of the synoptic conditions for this convective case is provided in section 3. The influence on precipitation and precipitation related variables is shown in section 4.1. An estimate on the model uncertainty based on the CTRL-ensemble is calculated in section 4.2. With the given uncertainty range, the significance of changes in precipitation caused by changes in soil moisture compared to the model spread is assessed in section 4.3, and systematics in the soil moisture impact are investigated in section 4.4.

## 2 Modelling approach

### 2.1 Numerical setup

We simulated the convectively induced precipitation on 3 August 2012 over the area around Hamburg (Germany), using the non-hydrostatic model COSMO (Version 4.22, Schättler et al., 2009) with a horizontal resolution of $0.1°$ ($\approx 1\,km$) for a simulation period of 24 hours. The chosen domain covers $400 \times 450$ grid points over Northern Germany (Fig. 1). 50 vertical hybrid Gal-Chen levels range from the surface up to a height of $22\,km$. The lowest level has a vertical resolution of $20\,m$. Boundary and initial conditions are provided by the COSMO operational analysis with a resolution of $2.8\,km$.

The horizontal resolution of approximately $1\,km$ allows for an explicit representation of deep convection, and thus provides much more accurate simulations of convective precipitation than resolutions that require convection parametrizations (Leutwyler et al., 2016, and references therein). Shallow convection is parametrized using the Tiedke Scheme (Tiedtke, 1989). Land surface processes are calculated by the interactive soil and vegetation model TERRA-ML, and coupled to the atmospheric module (Doms et al., 2011). The coupled soil model includes seven soil levels from the surface down to a depth of $14.58\,m$

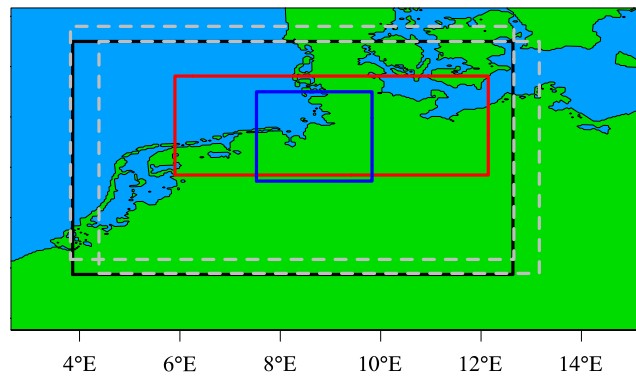

**Figure 1.** Model domain over Northern Germany given by the black rectangle for the CTRL run. Dashed gray rectangles describe the model domain shifted by 30 grid points to the north and to the east, respectively. The two analysis areas are marked with red and blue rectangles, hereafter referred to as area "red" and "blue", respectively.

with the uppermost layer having a depth of 5 mm.

## 2.2 Soil moisture experiments

To address the potential effect of soil moisture amount and local distribution on precipitation, the soil moisture content provided
in the initial conditions was modified (Table 1). Two types of changes in the soil moisture field were applied: extreme artificial changes and modifications in a physically feasible, realistic range (Fig. 2). As an extreme modification, the total drying of the soil is implemented by setting the soil moisture content to zero (Fig. 2c). Soil moisture increase is achieved by an increase of 50% (Fig. 2d) in all soil layers. These changes are first applied over all land points in the model domain ($DRY_a$ and $MOI_a$, Table 1) and second over land points in the domain framed in blue in Fig. 2d ($DRY_p$ and $MOI_p$, Table 1). Another artificial
modification is achieved by redistributing the soil moisture into four alternating bands (BAND) with 50% increased and reduced soil moisture (Fig. 2b). A large range of possible soil moisture effects is covered with these modifications. More realistic and less intense modifications are implemented by replacing the original soil moisture pattern by a real pattern of a different day (Fig. 2e). For this purpose, soil moisture fields from 19 July 2012 and 20 August 2012 are used (Fig. 3). On August 20, the soil moisture content in the uppermost soil layer (5 mm) is around 1.2 mm [$H_2O$] averaged over all land grid points in the model
domain which is 0.3 mm lower than on the simulated day (3 August 2012). On 19 July 2012 soil moisture content was slightly

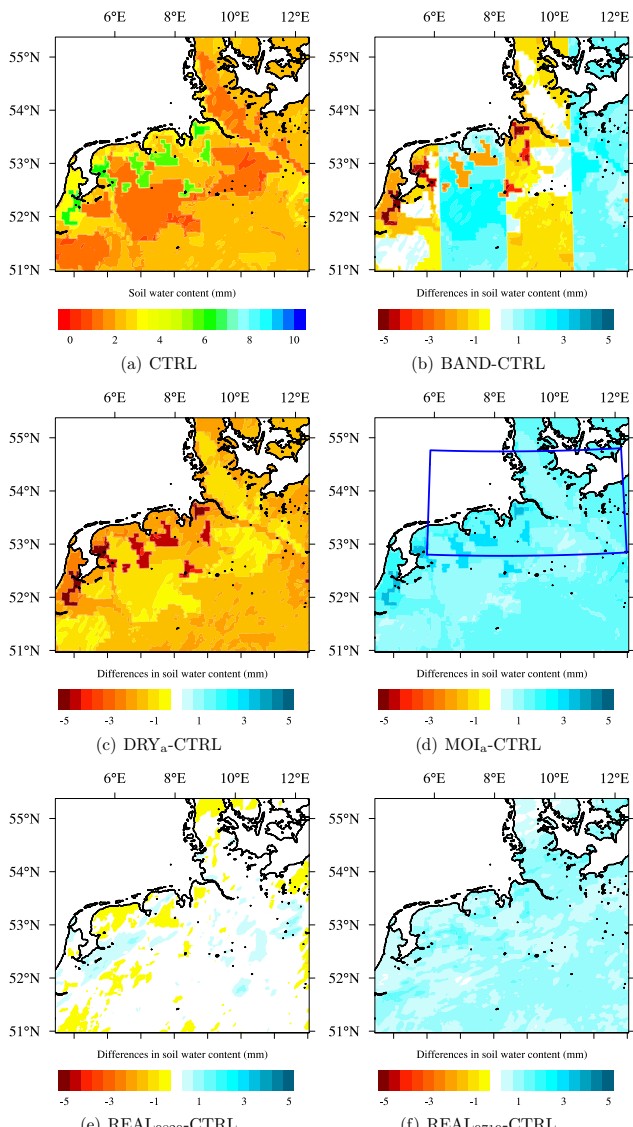

**Figure 2.** (a) Soil moisture for CTRL run and differences between CTRL run and (b) BAND run, (c) $DRY_a$ run, (d) $MOI_a$ run, (e) $REAL_{0820}$ run and (f) $REAL_{0719}$ run in the uppermost soil layer. Blue rectangle indicates the region where soil moisture was changed in $DRY_p$ run and $MOI_p$ run.

below the 50%-artificial increase (Fig. 3).

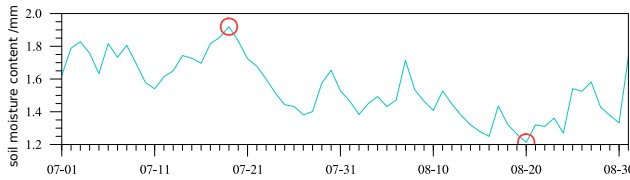

**Figure 3.** Time series for soil moisture content in the uppermost soil level averaged over the analysis domain "red". Red circles indicate the soil moisture values, which were used to perform simulations with soil moisture from another day.

**Table 1.** Model simulations with modified soil moisture (SM). Simulations are named by the applied soil moisture modification and with $a$ for whole model domain and $p$ for modification in a sub-domain (partly). Simulations with additional random changes caused by spatial shifting of the domain are denoted with $ii$ and $jj$, which represents the number of grid points by which the model domain is shifted (For details see Table 2).

| reference simulation for ensemble | Characteristics | | ensemble generation | |
| --- | --- | --- | --- | --- |
| | modification | area | | |
| CTRL | | | CTRL-LOC$ii\,jj$ | TIME$tt$ |
| DRY$_a$ | dry out | whole model domain | DRY$_a$-LOC$ii\,jj$ | |
| DRY$_p$ | dry out | area "red" | DRY$_p$-LOC$ii\,jj$ | |
| MOI$_a$ | 50% increased SM | whole model domain | MOI$_a$-LOC$ii\,jj$ | |
| MOI$_p$ | 50% increased SM | area "red" | MOI$_p$-LOC$ii\,jj$ | |
| BAND | four bands | whole model domain | BAND-LOC$ii\,jj$ | |
| REAL$_{0820}$ | SM from 20.08.12 | whole model domain | REAL$_{0820}$-LOC$ii\,jj$ | |
| REAL$_{0719}$ | SM from 19.07.12 | whole model domain | REAL$_{0719}$-LOC$ii\,jj$ | |

## 2.3 Ensemble approach

To quantify the relevance of the results from the soil moisture modifications, the model uncertainty and variability is estimated with a novel and simple approach. Perturbations are implemented in the simulations by shifting the domain boundaries by ten to 30 grid points north- and eastwards (Table 2, Fig. 1). These perturbations allow for estimating the uncertainty caused

5  by the chaotic behavior of the atmospheric system and are superimposed on all systematic and physical changes caused by soil moisture perturbations. This method conserves the structure of all meteorological input fields and does not create errors on a scale that can interact with the analyzed processes e.g. by creating small-scale secondary circulations. Furthermore, shifting start times of the simulations (Hohenegger and Schär, 2007) provide an additional degree of uncertainty with the same advantages as the domain shift. A time shift of one to six hours is also applied to the CTRL run to allow for a fair comparison

10  of uncertainty estimates. The ensemble, further called the CTRL-ensemble, comprises 17 independent model simulations,

**Table 2.** Uncertainty ensemble with randomly changed model simulations by model domain shifting (LOC) and the number of shifted grid points, and caused by shifting the model start time (TIME). The time shift is given in hours. The lower left (LL) corner of the simulation domains is given in geographical (rotated) coordinates with the north pole being shifted to 40° N and -170° E.

| run | LL corner in ° N | LL corner in ° E | starttime (UTC) |
| --- | --- | --- | --- |
| CTRL | 50.87 (1.0) | 15.55 (3.5) | 0:00 |
| LOC 00 10 | 50.97 (1.1) | 15.56 (3.5) | 0:00 |
| LOC 00 20 | 51.07 (1.2) | 15.57 (3.5) | 0:00 |
| LOC 00 30 | 51.17 (1.3) | 15.59 (3.5) | 0:00 |
| LOC 10 00 | 50.88 (1.0) | 15.39 (3.4) | 0:00 |
| LOC 10 10 | 50.98 (1.1) | 15.40 (3.4) | 0:00 |
| LOC 10 20 | 51.08 (1.2) | 15.42 (3.4) | 0:00 |
| LOC 20 00 | 50.89 (1.0) | 15.23 (3.3) | 0:00 |
| LOC 20 10 | 50.98 (1.1) | 15.25 (3.3) | 0:00 |
| LOC 20 20 | 51.08 (1.2) | 15.26 (3.3) | 0:00 |
| LOC 30 00 | 50.89 (1.0) | 15.08 (3.2) | 0:00 |
| TIME 01 | 50.87 (1.0) | 15.55 (3.5) | 1:00 |
| TIME 02 | 50.87 (1.0) | 15.55 (3.5) | 2:00 |
| TIME 03 | 50.87 (1.0) | 15.55 (3.5) | 3:00 |
| TIME 04 | 50.87 (1.0) | 15.55 (3.5) | 4:00 |
| TIME 05 | 50.87 (1.0) | 15.55 (3.5) | 5:00 |
| TIME 06 | 50.87 (1.0) | 15.55 (3.5) | 6:00 |

including the reference simulation (CTRL-run), to estimate the uncertainty. This ensemble generating approach, including shifted model domain, is applied to each simulation with modified soil moisture patterns (Table 2).

## 3    Convective case study and the effect of soil moisture

The chosen convective case of 3 August 2012 is characterized by a low pressure system over the northern Atlantic, west of Great Britain (Fig. 4). The associated cold front moved across Germany and resulted in heavy precipitation over Poland where air masses converged. Adjacent to the major precipitation events in the east, another local precipitation cell developed close to Hamburg where a slight enhancement in convective available potential energy (CAPE) values and high clouds were observed (Fig. 4). This strong local precipitation cell was detected by the rain radars over Northern Germany at 14:11 UTC with rain rates between 10 to 100 $\mathrm{mm\,h^{-1}}$ (Fig. 5). The COSMO simulations showed maximum values of 12 $\mathrm{mm\,h^{-1}}$ between 13:00 and 17:00 UTC. The simulated precipitation onset is around 10:00 UTC (Fig. 7a). Before the onset of precipitation, high CAPE

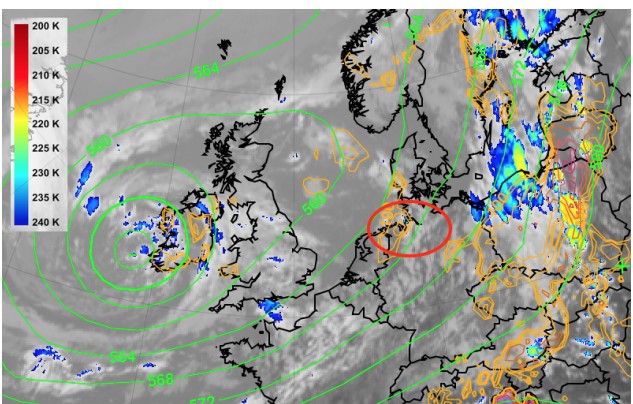

**Figure 4.** EUMeTrain infra-red satellite image of Europe on 3 August 2012 at 12:00 UTC. Colors are cloud top temperatures showing high clouds. Green contour lines are geopotential height at $500\,\text{hPa}$ and orange contour lines are CAPE values provided from ECMWF NWP (www.eumetrain.org). The area of interest is marked with a red circle. Note the CAPE values within the marked area and the high clouds, which correspond to the intense precipitation.

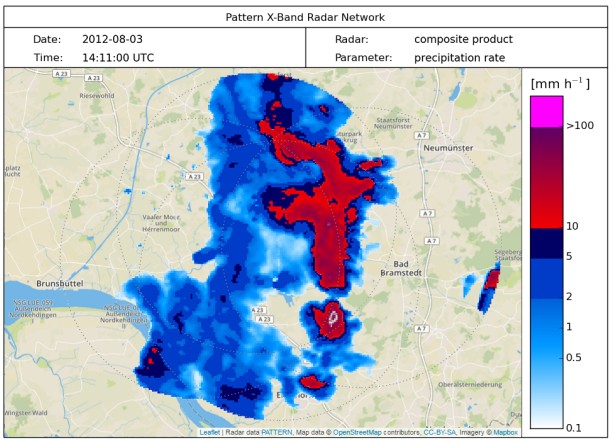

**Figure 5.** Radar composite from high resolution radars (Lengfeld et al., 2014) showing the precipitation rate over Northern Germany on 3 August 2012 at 14:11 UTC.

values confirm the precipitation's convective nature (Fig. 7b). High CAPE values indicate the development of strong convective precipitation presuming that CIN can be exceeded.

## 4 Results

### 4.1 Soil moisture influence on convection-related variables

While the passing front is the main mechanism for the lifting of air masses in the performed simulations, soil moisture is important for the stability of the atmosphere and thus affects precipitation initially triggered by the synoptic system. In completely dry conditions ($DRY_a$ and $DRY_p$), the latent heat flux is zero and the sensible heat flux alone needs to balance the net radiation flux and the soil heat flux (Fig. 6). Without latent cooling, the temperatures at $2\,m$ altitude in the simulation with dry soil conditions ($DRY_a$ and $DRY_p$) are the highest, while dew point temperatures are the lowest (Fig. 6c and d). With more humidity in the atmosphere, less adiabatic cooling due to lifting is required for condensation. The resulting shift in the condensation level to lower altitudes can reduce CIN and increase CAPE. CIN only decreases in the first hours of the simulation before solar radiation heats the surface (Fig. 7). When the surface heats up in simulations with low soil moisture content ($DRY_a$ and $DRY_p$), CIN is continuously lower than in simulations with higher soil moisture content ($MOI_a$ and $MOI_p$). The further development of CIN is strongly affected by the feedback from precipitation.

The convective related quantities react systematically to changes in soil moisture amount (Fig. 6). However, the relation between CIN and soil moisture varies with the diurnal cycle. Convective precipitation is more likely with reduced CIN, but a low CIN is not associated with stronger precipitation. The strength of precipitation depends on CAPE. Thus, precipitation does not respond systematically to changes in soil moisture. The main reason for this unsystematic behavior mentioned in literature is the dependency of CIN on the soil moisture (Kalthoff et al., 2011), which is a measure for the probability of convection but is not directly related to precipitation intensity. Even though the changes in precipitation caused by changing soil moisture are more complex, precipitation is certainly influenced by the soil moisture. However, the changes in precipitation caused by modifications in soil moisture are often less significant than changes caused by synoptic forcing as will be demonstrated below.

### 4.2 Estimate of model uncertainties

An estimate of the model uncertainty is determined from the CTRL-ensemble. The assessment of this uncertainty is done statistically by using the SAL score (Wernli et al., 2008), which assigns values for differences in structure, $S$, amplitude, $A$, and location, $L$, between precipitation patterns at every single output time step (15min). These three parameters of the SAL score are briefly introduced in the following.

Amplitude, $A$, describes the differences of precipitation amount over the whole analyzed domain:

$$A = \frac{D(R_{dif}) - D(R_{ref})}{0.5[D(R_{dif}) + D(R_{ref})]}. \tag{1}$$

The precipitation amount averaged over the whole domain $D(R)$ is:

$$D(R) = \frac{1}{N_{GP}} \sum_{(i,j)\in\epsilon} R_{ij}, \tag{2}$$

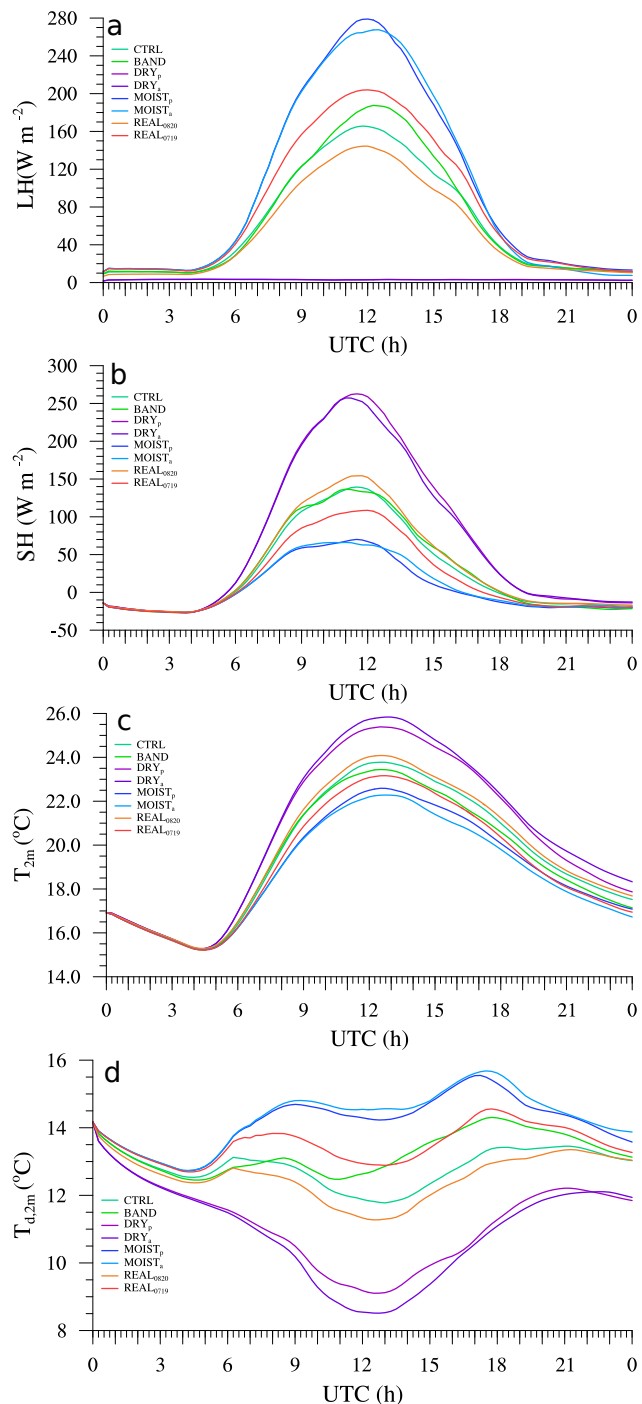

**Figure 6.** Time series of (a) latent heat fluxes, $LH$, (b) sensible heat fluxes, $SH$, (c) 2m-temperature and (d) dew point temperature, $T_d$ in the reference simulations of the ensembles with different soil moisture modifications for 3 August 2013 averaged over area "red".

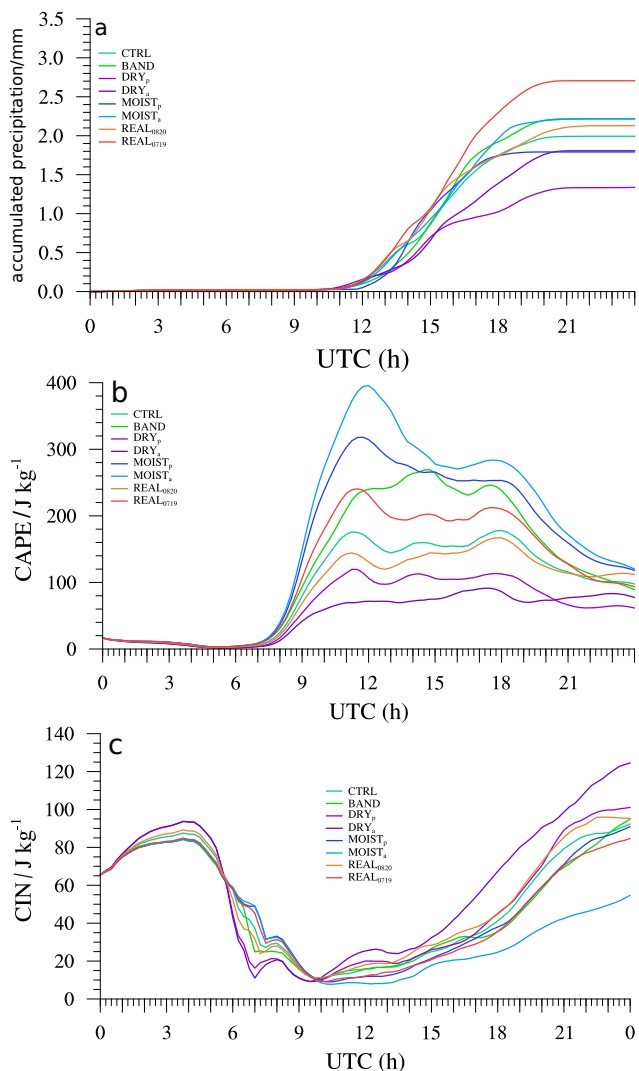

**Figure 7.** Time series of (a) accumulated precipitation, (b) CAPE and (c) CIN for different model simulations for 3 August 2013 averaged over "red" area.

where $R_{ij}$ is the precipitation rate at the grid point with indices i, j, and $N_{GP}$ the number of all grid points in the analyzed domain. The horizontal grid points are approximately equally spaced in the limited model domain. $D(R_{dif})$ denotes the averaged precipitation amount for shifted simulations and $D(R_{ref})$ the reference simulations (not shifted) in the CTRL-ensemble.

The location parameter, $L$, compares the location of precipitation in two model simulations in two steps. First, the normalized distance of the centers of mass, $\mathbf{x}(R)$, of the precipitation patterns in each model simulation is calculated:

$$L_1 = \frac{|\mathbf{x}(R_{dif}) - \mathbf{x}(R_{ref})|}{d}, \tag{3}$$

where $d$ denotes the maximum possible distance within the analyzed domain. Secondly the distances from the center of mass of all $M$ individual cells, $\mathbf{x}_n$, to the center of mass for the whole precipitation field, $\mathbf{x}$, are calculated as:

$$r(R) = \frac{\sum_{n=1}^{M} R_n |\mathbf{x} - \mathbf{x}_n|}{\sum_{n=1}^{M} R_n} \tag{4}$$

The distances resulting from the reference simulation and shifted simulations are then compared:

$$L_2 = 2 \left[ \frac{|r(R_{dif}) - r(R_{ref})|}{d} \right]. \tag{5}$$

Afterwards, both components of $L$ are added.

The structure component, $S$, indicates whether the precipitation patterns tend to more convective precipitation with small but more peaked rain objects or to shallow precipitation with larger objects, but weaker precipitation intensity. A volume, $V(R)$, is calculated by dividing the precipitation sum for a cell n, $\mathcal{R}_{ij}$, calculated over the $\epsilon$ grid cells of n, by the maximum precipitation of this cell, $R_n^{max}$:

$$V_n = \sum_{(i,j) \in \epsilon} \frac{\mathcal{R}_{ij}}{R_n^{max}}, \tag{6}$$

$$V(R) = \frac{\sum_{n=1}^{M} R_n V_n}{\sum_{n=1}^{M} R_n}. \tag{7}$$

With the volume, $V(R)$, over all precipitation cells, $M$, the structure component can be calculated similarly to Eq. (1):

$$S = \frac{V(R_{dif}) - V(R_{ref})}{0.5[V(R_{dif}) + V(R_{ref})]}. \tag{8}$$

For more detailed information on the SAL-score see Wernli et al. (2008).

The simulations are compared for the period from 10:00 UTC to 18:00 UTC, covering the precipitation event. Simulations with a shifted model start or domain are compared to the CTRL run. Amplitudes, $A$, are positive or negative depending on which run is used as reference (ref) or as a comparison simulation (dif). To avoid an uncertainty range shifted toward one sign, all comparisons are additionally performed after swapping ref and dif runs, providing a symmetric uncertainty distribution. The uncertainty estimate (Fig. 8) encompasses a sample of 122 (permutations of simulation couples) $\times$ 32 (time steps) values, although not all of them are independent of each other.

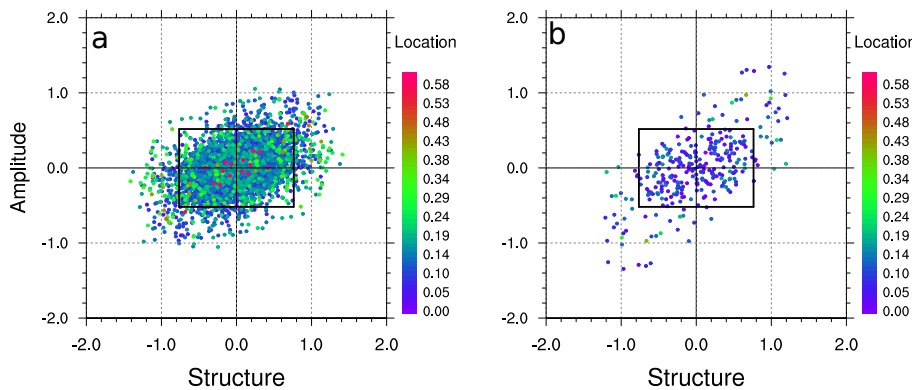

**Figure 8.** SAL results for the CTRL-uncertainty-ensemble between 10:00 UTC and 18:00 UTC separated for uncertainty generated by the spatially shifted model domain (a) and by the delayed model start (b). Structure is represented on the x-axis, amplitude on y-axis and location by marker color. Each marker shows a comparison between two model simulations at a single time step. Simulations with shifted model domain are represented by filled dots and simulations with shifted model start time by rectangles. The grey rectangle delimits the region between the 5% and 95% percentiles in $S$ and $L$ amplitude.

As parameters $A$ and $S$ are correlated (Fig. 8), a reduction in precipitation amplitude is related to too small and/or peaked precipitation objects, whereas an increase in precipitation amount goes along with larger and/or shallower rain objects. The largest amplitude deviations between the different runs arise in the first hours of the analysis time from 10:00 to 11:30 UTC (Fig. 9). This coincides with the time of the onset of the precipitation event which differs in the different simulations (Fig. 7)

and therefore causes the largest uncertainties. The end of the precipitation event is not considered in this particular time range. A large shift in model start time leads to higher uncertainties (Fig. 9). Changes due to the spatially shifted model domain do not depend on the distance of the boundary shift. The deviations from CTRL for simulations with shifted boundaries are not caused by a direct change of physical parameters such as temperature. The differences emerge because the synoptic forcing at the lateral model boundary differs. Deviations are further caused by changes in the lower boundaries, such as changing areas

of sea- or land-cover. For example, a model domain with north- or westward shifted boundaries includes more grid points over sea surface. The simulation with the strongest westward shift (LOC3000) shows the largest changes in precipitation amplitude (Fig. 9). However, the strongest shift (30 km north in LOC3000) affects the precipitation amplitude less than a smaller shift (20 km north in LOC2000), which includes a smaller fraction of sea surface (Fig. 9).

To address the dependency of the SAL score on the chosen analysis area, two different analysis areas are chosen (Fig. 1).

The area framed in blue in Fig. 1 includes mainly small convective cells and the area framed in red includes the whole precipitation field. For these two analysis areas, two simulations are compared. We define the model uncertainty for this study as the range between the 5% and 95% percentiles for $S$ and $A$ and up to the 90% percentiles for $L$. According to this definition, the uncertainty range is $\pm0.77$ ($\pm0.86$) in $S$, $\pm0.54$ ($\pm0.69$) in $A$, and up to 0.20 (0.29) in $L$ for analysis area "red" ("blue"). Changes are considered significant when the response to the soil moisture modification is larger than the generated background

noise from spatially shifted model domains or delayed model start times. When comparing the CTRL run (Fig. 10a) and the

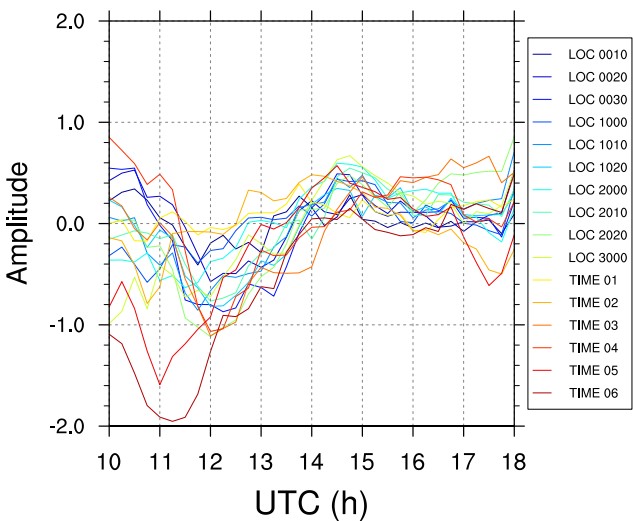

**Figure 9.** Amplitude values from Fig. 8 for comparisons to the CTRL run only, for single time steps.

simulation with shifted boundaries (Fig. 10b), differences in individual cells in the western part of the domain, partly over the North Sea, and in the structure of the large precipitation pattern in the eastern part of the domain become obvious. These differences are caused by shifting the boundary domain by ten grid points (10 km). The extreme modifications in soil moisture cause even more apparent differences in the precipitation patterns. The increase of soil moisture in either the whole domain or

5 in a sub-domain dramatically changes the location of the precipitation cells in the analyzed time steps (Fig. 10e and f). In the moist simulations ($MOI_a$ and $MOI_p$), the strongest precipitation occurs north-east of the Elbe estuary. This region is mainly free from precipitation in the dry simulations ($DRY_a$ and $DRY_p$) in Fig. 10c and d. The precipitation is simulated even further north-east of this particular region in the CTRL simulations (CTRL and LOC1000) in Fig. 10a and b. Moderate changes in soil moisture, e.g. when applying realistic moisture fields of a different day, result in smaller changes in precipitation. The general

pattern observed in the CTRL run remains the same in $REAL_{0820}$ and $REAL_{0719}$ (Fig. 10f and g).

## 4.3 Significant effects of soil moisture modification on precipitation

The large number of model simulations (a complete ensemble for each soil moisture modification) and the uncertainty estimate from section 4.2 allows for a quantitative evaluation of the significance of soil moisture influence on precipitation. Each

15 ensemble with modified soil moisture is compared to the CTRL uncertainty ensemble by comparing ensemble members with the same spatial shift of the model domain for each output time step applying the SAL score. Within every ensemble the SAL-values are divided into those that exceed the uncertainty range given by the blue rectangle in Fig. 11 and those that lay within this range. The uncertainty range is estimated from the uncertainty-ensemble. The percentage, $p$, of values exceeding the uncertainty range is calculated to decide whether soil moisture modification leads to significant changes in precipitation

(bold $p$ values in Table 3). Changes caused by a soil moisture modification are considered as significant if more than 10% of the values exceed the uncertainty range. The threshold is set to 10% because the uncertainty range is determined by considering the range between the 5 and 95 percentiles, which leaves 10% probability that the exceeding value is still caused by model uncertainty.

The change in $S$ of precipitation caused by soil moisture modification in the $DRY_p$-ensemble exceeds the uncertainty range in only 5% of all cases (Fig. 11a and Table 3). For both scores, $S$ and $A$, the percentage of exceeding values lies beneath the 10% threshold. Therefore, precipitation does not respond significantly to $DRY_p$ modifications in terms of $A$ and $S$ except for the parameter $L$ in area "red" (Table 3). In contrast, the soil moisture reduction in the whole domain ($DRY_a$-ensemble) affects the precipitation significantly (Fig. 11b). More than 50% of $A$ values exceed the uncertainty range, in some cases with values

for $A$ down to -1.8. For $S$, only 11% of the values exceed the uncertainty range. Nevertheless, this is enough to be classified as a significant impact. The soil moisture increase in a sub-domain only ($MOI_p$-ensemble) results in significant changes in precipitation (Fig. 11c). As already seen in the $DRY_a$-ensemble, the modification over the whole domain results in an even stronger precipitation response.

The redistribution of soil moisture (BAND-ensemble) does not lead to a significant effect (Fig. 11e), except for $L$ in area "blue".

This modification changes the heterogeneity of the soil moisture by reducing small-scale structures, but induces stronger variations on a large scale. Thus, secondary circulations can develop on a different scale. This is in accordance with Adler et al. (2011) and Kang and Bryan (2011) who both found an influence of the redistribution of soil moisture on the location of convective initiation. Therefore, area "blue", mainly containing small convective cells, is influenced to a greater extent than area "red", which has a large advected precipitation band.

Even slight modifications of soil moisture, as Klüpfel et al. (2011) achieved by using different initializations of soil moisture, lead to different precipitation patterns. In the present study, using soil moisture from a different day also changes precipitation in Fig. 11f. But these changes do not exceed the model uncertainty in more than 10% of all values in the present case. Accordingly, physically realistic changes in soil moisture lead to changes in precipitation not larger than changes that can also be caused by choosing a slightly different model setup.

**4.4  Systematic behavior to soil moisture changes**

Having determined the significance of the strength of changes in precipitation, this section deals with the systematics of changes. Significant changes do not necessarily imply systematic changes. Changes in $L$ are not analyzed as $L$ only describes the magnitude of the cell shift, but does not provide information on the direction of the shift.

While the value of $A$ is predominantly negative in $DRY_a$ (Fig. 11b), changes in $MOI_p$ (Fig. 11c) are significant, but random.

$S$ and $A$ are not correlated in any of the soil moisture experiments (Fig. 11) in contrast to the CTRL ensemble. To carve out any systematic effect, the averaged values of $A$ and $S$ are compared to the average of the uncertainty-ensemble. The sample for the SAL results for the uncertainty-ensemble is symmetric and therefore the average over all values is zero. A significant difference of the averaged values from zero hints at the systematics. Whether the averaged values differ significantly from zero

**Table 3.** Percentages $(p_S, p_A, p_L)$ of values $S$, $A$ and $L$. Uncertainty is in a range from $[-0.767, 0.767]([-0.857, 0.857])$ in structure, $[-0.538, 0.538]([-0.690, 0.690])$ in amplitude and $0.200(0.288)$ in location for analysis area "red" ("blue"). Bold values exceed model uncertainties in more than 10% of the cases. Averaged values and their deviations $(\overline{S} \pm \hat{\sigma}^2, \overline{A} \pm \hat{\sigma}^2)$ are also listed. Bold values are mean values that differ significantly from the mean of the uncertainty-ensemble after Eq. (9) for confidence interval of 90%.

| Ensemble | Structure | | Amplitude | | Location |
|---|---|---|---|---|---|
| | $p_S$ | $\overline{S} \pm \hat{\sigma}^2$ | $p_A$ | $\overline{A} \pm \hat{\sigma}^2$ | $p_L$ |
| analysis area "red" | | | | | |
| $DRY_p$ | 5.79 | $0.02 \pm 0.0034$ | 3.58 | $-0.13 \pm 0.0011$ | **25.34** |
| $DRY_a$ | **23.14** | $\mathbf{0.30 \pm 0.0063}$ | **22.31** | $-0.26 \pm 0.0023$ | **53.72** |
| $MOI_p$ | **15.98** | $-0.12 \pm 0.0051$ | **18.73** | $-0.05 \pm 0.0042$ | 8.26 |
| $MOI_a$ | 9.92 | $-0.10 \pm 0.0043$ | **23.42** | $\mathbf{-0.18 \pm 0.0043}$ | **26.72** |
| BAND | 3.03 | $-0.04 \pm 0.0025$ | 0.55 | $0.00 \pm 0.0001$ | 6.61 |
| $REAL_{0820}$ | 3.31 | $-0.03 \pm 0.0022$ | 0.28 | $-0.02 \pm 0.0006$ | 0.55 |
| $REAL_{0719}$ | 2.48 | $0.05 \pm 0.0024$ | 1.65 | $0.09 \pm 0.0008$ | 1.65 |
| analysis area "blue" | | | | | |
| $DRY_p$ | 4.85 | $-0.10 \pm 0.0033$ | 9.39 | $\mathbf{-0.29 \pm 0.0091}$ | 5.76 |
| $DRY_a$ | **11.82** | $0.08 \pm 0.0058$ | **51.21** | $\mathbf{-0.60 \pm 0.0068}$ | **30.61** |
| $MOI_p$ | **14.85** | $-0.19 \pm 0.0053$ | **12.12** | $0.00 \pm 0.0039$ | **12.42** |
| $MOI_a$ | **15.76** | $\mathbf{-0.27 \pm 0.0058}$ | **19.70** | $-0.28 \pm 0.0044$ | **27.58** |
| BAND | 7.27 | $-0.10 \pm 0.0045$ | 0.91 | $0.07 \pm 0.0014$ | **21.52** |
| $REAL_{0820}$ | 0.91 | $-0.04 \pm 0.0026$ | 0.30 | $-0.03 \pm 0.0007$ | 1.21 |
| $REAL_{0719}$ | 1.21 | $-0.01 \pm 0.0026$ | 0.30 | $0.07 \pm 0.0010$ | 1.21 |

is tested statistically by:

$$\widehat{z}_{\text{sys}} = \frac{\overline{x}_1 - \overline{x}_2 - E[\overline{x}_1 - \overline{x}_2]}{\sqrt{\hat{\sigma}^2[\overline{x}_1 - \overline{x}_2]}}. \tag{9}$$

$\overline{x}_1$ and $\overline{x}_2$ denote the averaged values of $S$ or $A$ for the two compared simulations, $E[\overline{x}_1 - \overline{x}_2]$ is the expected value for the differences between the two simulations and is expected to be zero for the null hypothesis and $\hat{\sigma}^2[\overline{x}_1 - \overline{x}_2]$ is the variance of the averages.

The average of $S$ differs significantly from zero only for the ensembles $DRY_a$ and $MOI_a$ in the analysis domain "red" and "blue", respectively (Table 3). The change in precipitation amount indicated by $A$, from the two ensembles with reduced soil moisture ($DRY_a$ and $DRY_p$) compared to the control ensembles deviates significantly from zero towards negative values. Thus precipitation is reduced systematically in simulations with reduced soil moisture. This result is robust for both analyzed areas. Contrarily to the simulation with decreased soil moisture, a systematic reduction in precipitation is also found in simulations with increased soil moisture over the whole domain ($MOI_a$) independent of the particular analysis area. The positive feedback

activated by reduced soil moisture is in line with a case study by Barthlott and Kalthoff (2011). However, increased soil moisture amount can lead to an increase or decrease in precipitation, dependent on the strength of the increase. In contrast, Cheng and Cotton (2004); Ek and Holtslag (2004); Martin and Xue (2006); Hohenegger et al. (2009); Weverberg et al. (2010) found a negative feedback in convection resolving simulations.

The strength of the deviation depends on the strength of the modification. While a area limited increase in soil moisture does not lead to systematic changes, an overall increase has a systematic effect. The effect of dry soil exceeds the effect of soil moisture increase and shows systematic effects for both implementations (drying the entire domain or only parts). The effects are more strongly for overall modifications. Comparing the results for both regions, the averaged differences calculated for region "blue" exceed those of region "red". That is because region "blue" covers more locally initiated convective cells, which are affected

stronger by soil moisture than advected precipitation cells which are influenced stronger by the large scale dynamics.

## 5 Conclusions

In the present case study, we carried out seven separate ensembles for different perturbations in soil moisture amount and soil moisture pattern. Each ensemble was composed of ten variations of the model domain. The soil moisture perturbations include both strong artificial changes by drying and wetting the model domain, and realistic changes implemented by replacing the

15 initial soil moisture field with real soil moisture patterns of a different day. The ensembles are generated by conducting the simulation over slightly different model domains, each one created by shifting the domain location by 10 to 30 km in order to change the boundary conditions. These changes cause a large spread in the investigated case because, even though precipitation at that day was of convective nature (3 August 2012), additionally, there was strong synoptic forcing by a low pressure system over the Atlantic. Conclusively, only one (three) ensemble in area "red" ("blue") shows significant changes caused by modified

soil moisture amount in intensity, local distribution and amount of convective precipitation which are assessed using the SAL-score. The amplitude, a measure of the difference in the amount of precipitation, is mostly systematically reduced within an ensemble. No overall systematics were found because both wetting and drying of soil can result in reduced precipitation amount. The structure, which describes the spatial variability of the precipitation field, can either be increased or decreased at different times and in different ensemble members. This might be explained by a delayed onset of precipitation caused by soil

moisture modifications. A local displacement in the precipitation cells is found for three (four) out of five artificially changed soil moisture patterns in analysis region "red" ("blue"). The changes in precipitation for the simulations with realistic soil moisture patterns are not significant.

A limitation of this study is the restriction to a single case study. Thus, no generally valid conclusions can be drawn. However, this study presents a proof of concept and should be further conducted with cases that are less affected by frontal systems. In

those cases, a stronger influence of soil moisture on precipitation properties may be expected. A second limitation is caused by the dependency of the results on the chosen analysis area, which in turn shows the complexity of the results. Furthermore, the uncertainty estimate depends on synoptic forcing and size of the model domain. The ensemble spread might become smaller with weaker synoptic forcing and with a larger model domain. However, a smaller model spread would strengthen the

importance of soil moisture influence as to be expected in these cases.

In summary, we could prove our concept on creating a sufficiently large model spread by shifting the model boundaries. This ensemble generation technique does not generate any patterns in the initial conditions, which could cause scale interaction and secondary circulations. Such an estimate of the model spread is necessary in soil moisture studies to separate the response to soil moisture changes from inherent forecast uncertainty at deep convection resolving grid spacing. We further showed that in synoptically driven situation, the effect of soil moisture remains uncertain and further investigation is necessary.

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

## Data availability

Model output data are stored at DKRZ computers and can be provided on demand.

*Acknowledgements.* We would like to thank the COSMO consortium for access to the code, and the German weather service (DWD) for providing analysis data, Deutsches Klimarechenzentrum (DKRZ) for providing a simulation platform and Heini Wernli and Markus Zimmer for the SAL analysis code. Thanks to Will Ball, Claire Merker, Katharina Lengfeld and Ben Schlifke for proofreading. We thank the anonymous reviewers for their fruitful comments to improve this manuscript and the editor for his support and patience.

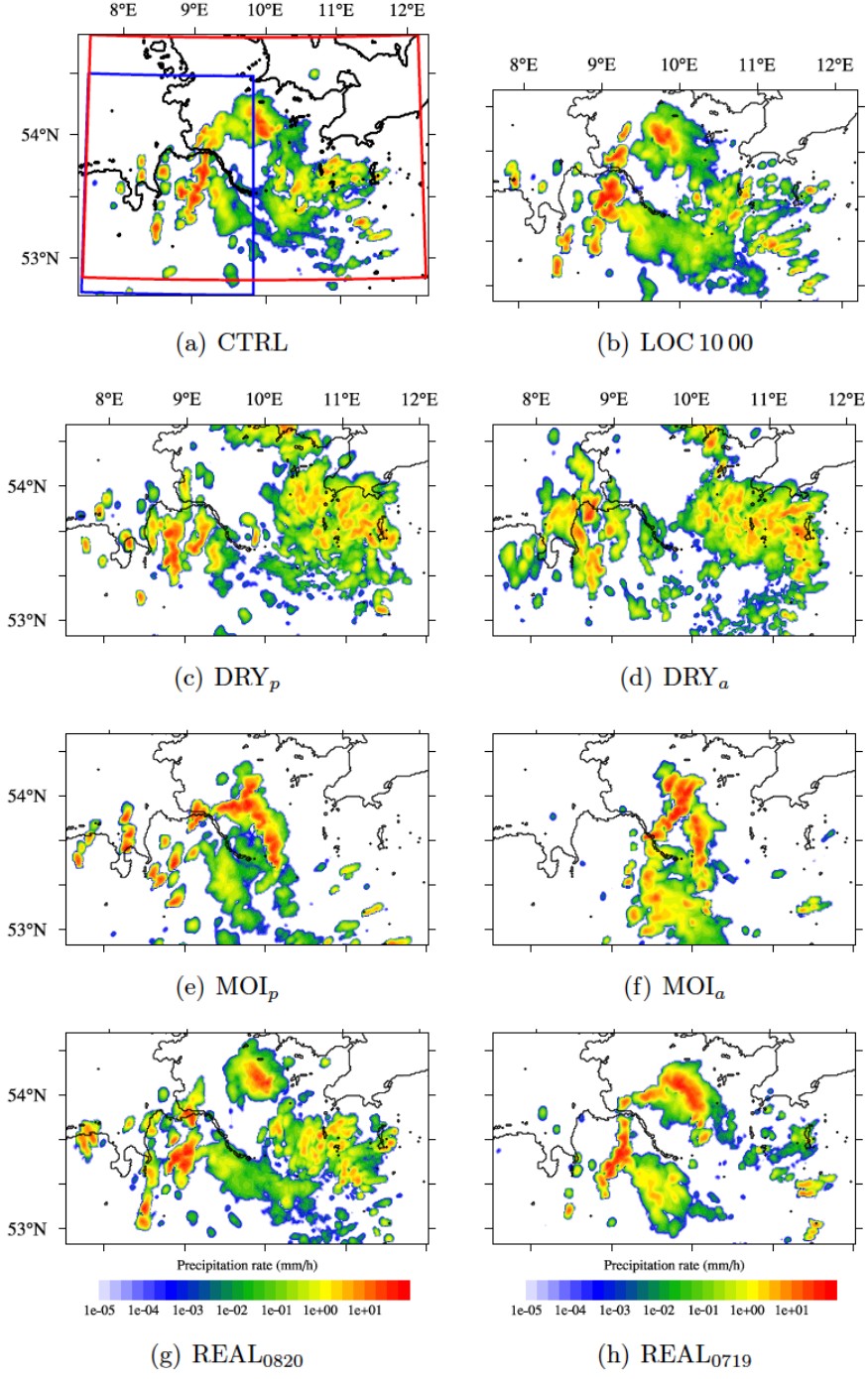

**Figure 10.** Precipitation rate at 14:45 UTC for (a) CTRL run , (b) LOC 10 00 and (c-h) different soil moisture modified simulations.

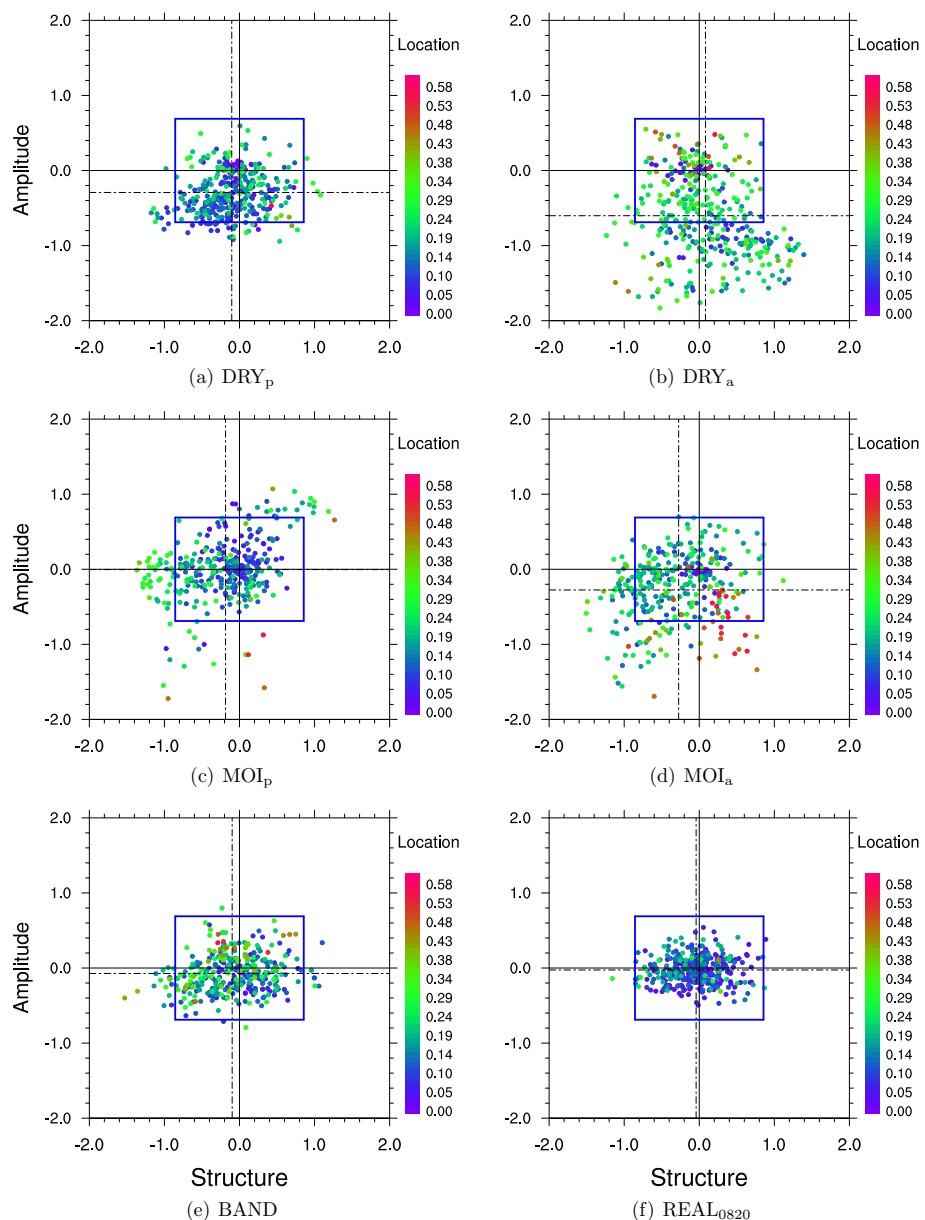

**Figure 11.** SAL scatter plot: Comparison of ensembles (a) $DRY_p$, (b) $DRY_a$, (c) $MOI_p$, (d) $MOI_a$, (e) BAND and (f) $MOI_{0820}$ with the uncertainty-ensemble for area "blue". Dashed lines represent the averages.