# Peer review of "Assessing the uncertainty of soil moisture impacts on convective precipitation using a new ensemble approach"

_Atmospheric Chemistry and Physics, 2016_

## Referee Comment (RC1) · Anonymous Referee #1 · 3 Mar 2017

General comments
This paper aims at assessing the soil moisture-precipitation feedback for one case in northern Germany with numerical simulations using the COSMO model. Besides a control run, several sensitivity simulations were performed with reduced/increased soil moisture, with a banded soil moisture distribution, and realistic soil mositure values from other days. These model runs are compared to runs with shifted model domains and different initialisation times to distinguish between random changes in precipitation and changes that result from soil moisture.

Although the concept itself is promising, I have a number of major concerns with respect to the representativeness of the results and the applied method. Some of my concerns are probably caused by clumsy English phrases that will need to be addressed before the paper is suitable for publication. I expand on some of these concerns below and outline additional major and minor comments. My recommendation is major revisions.

1. I like the concept of distinguishing between random changes in precipitation and changes that result from soil moisture, but I have some concerns about the representativeness of the results. So far, I am not convinced that the simulation of one day and the evaluation on two comparably small evaluation domains is the right concept. For a more robust conclusion, more cases are necessary and the analysis of one evaluation area alone might be more meaningful. As the authors already mention in the Conclusions, further case studies are needed. If this paper is intended to be a proof-of-concept, the authors should clearly state that in the manuscript and be cautious with any general conclusions.

2. My main criticism is due to the fact that nothing is being said about the physical processes that are responsible for these differences. The different model runs are compared to each other with the SAL method, but reasons for the differences remain unclear. As the paper is comparably short, I recommend to add a section on the physical processes responsible for the differences. For example, domain-averaged time series of convection-related parameters could be shown here.

3. When performing a sensitivity study, the control run has to be evaluated first to assure that it serves as a good basis for the sensitivity runs. I believe you need to insert a subsection on the synoptic controls, the observed precipitation and the results of the control run.

4. In many operational forecasting centers, soil moisture is already perturbed in their ensemble prediction systems. Some information about that and most importantly, the differences to the method used in this paper, should be added to the manuscript.

Specific comments

1. P1, L2: What do you mean by model uncertainty? Please clarify. Again, later on: "Only drastic soil moisture changes can exhibit the model uncertainties..." Probably you mean similar uncertainties as in other ensemble systems where e. g. stochastic perturbations are inserted, tuning parameters changed, or different initial and boundary data from another model are used. This has to be made clearer at several locations in the manuscript.

2. P1, L4: "...but the systematic behaviour is still complex..." Up to now, there is no consent about the existence of a systematic relationship of soil moisture to precipitation. I would rather write: "...but the response of precipitation to soil moisture changes is still complex..."

3. P1, L6: Some details about the ensemble approach used in this work should be given here.

4. P1, L23: Surface temperatures are dependant on the sensible heat flux, not the latent heat flux. Please rephrase.

5. P2, L1: You mean the water content of air? Then it's probably better to write: "Secondly, soil moisture strongly influences the low-level humidity via the latent heat flux."

6. P2, L5: "...react on the soil moisture." Better: "...depend on soil moisture due to its effects on low-level temperature and humidity."

7. P2, L16: What do you mean with the synergy of soil moisture-precipitation feedbacks?

8. P3, L3: Is shallow convection still parameterized? Which COSMO version do you use?

9. P3: Is the total drying of the soil the respective permanent wilting point? With the 50% increase in soil moisture, did you assure that you don't have larger values than the porosity allows? You state in the manuscript that you want to show the full range of soil moisture influence. So why did you use just a 50% increase and not the maximum value possible for the respective soil type? Did you change all levels in the soil in the same way? Did you make the changes at the model initialization time?

10. P4, Figure 2: This figure is too pixelated, the text is hardly readable.

    Concerning your band pattern: Does the soil moisture changes from 1 grid point to the other or is there a smoother transition over a couple of grid points? Do these strong gradients introduce any thermal direct circulations?

11. P5, L4: Which moist simulation do you refer to? I don't agree with the statement that in the moist simulation, precipitation occurs mainly at places that are free of precipitation in the CNTRL run. At least, I don't see that in Figure 3.

12. Figure 3: Instead of showing one time of day, a 24-h accumulated precipitation would be much more meaningful. Soil moisture may also influence the timing of cloud formation, so one snapshot might not be enough to show the overall effect. In addition, time series of domain-averaged precipitation should be shown as well.

13. P7, L25: Random perturbations are introduced by shifting the domain boundaries. Please explain in more detail, why you consider this as random perturbations. One way to prove that would be to insert stochastic perturbations e. g. in the initial temperature field. The authors should comment on that.

14. P8, L9: I don't understand what is meant with "reversed direction", please clarify.

15. P13, L13: "... show a positive feedback for decreased soil moisture." This is misleading, in their paper they find a positive feedback for increasing soil moisture but only for relatively dry soils. Please rephrase.

Technical corrections

1. I suggest to change the title to: Assessing the uncertainty of soil-moisture impact**s** on convective precipitation by an ensemble approach.

2. P2, L16: "...over complex terrain", orographic is not needed when using complex terrain

3. P5, L10: ...will be introduce**d**...

4. P6, L2: You name a modified model simulation **mod**, why not name the control run **ctrl** instead of **comp**?

5. P7, L6: fore should be for

6. P11, L2: convective inition should be convection initiation

7. P12, table caption: ...mean of the uncertainty-ensemble**after** after...

8. References: Please use the abbreviations used from the respective journals:
   Atmos. Res.
   J. Atmos. Sci
   Meteorol. Z.
   J. Hydrometeor.
   Q. J. R. Meteorol. Soc.
   Geophys. Res. Lett.
   J. Climate
   Mon. Wea. Rev.

[Figure]

Geosci. Model Dev.
Boundary-Layer Meteorol.
Bull. Amer. Meteor. Soc.
Atmos. Chem. Phys.

9. P13, L31: Something is wrong with the last sentence, please rephrase.

10. P14, L9: The expression "systematics of precipitation" is unusual, perhaps better: "especially concerning the existence and strength of the soil moisture-precipitation feedback."

11. P15, L11: Here are LaTeX commands in the link: `\T1` etc., please remove them.

12. P15, L20: Germany, not germany

13. P16, L13: Please remove the brackets from {E}, {GPU}, {COSMO}

14. P16, L21: Something is wrong with the entry for the pages: 407–430+341?

15. P16, L22: Muhlbauer, A.

16. P16, Reference Schättler et al.: The latest version of the COSMO user guide is from the year 2016, please update your entry.
* * *

---

## Referee Comment (RC2) · Anonymous Referee #2 · 11 May 2017

**Review on „Assessing the uncertaintiy of soil-moisture impact on convective precipitation by an ensemble approach" by O. Henneberg, F. Ament, and V. Grützun**

In this article, the authors evaluate the impact of different soil-moisture initializations on the simulation of convective precipitation with the COSMO model, using a set of ensemble simulations for one case study. These consist of 8 uncertainty ensembles based on one soil-moisture ensemble. The uncertainty ensembles consist of 11 simulations with a shifted model domain and in one case, on 6 additional simulations with a modified start time.

The underlying idea is that the uncertainty ensemble is a method to estimate model uncertainty, which is then used to assess the significance of different soil-moisture initializations. It is found that „only drastic soil moisture changes" can overcome the model uncertainties.

The idea to compare model uncertainty with soil-moisture induced uncertainty seems to be somehow neglected in recent literature and therefore the article is recommended for publication, even if it is not entirely clear if domain shifting can be regarded as a reliable measure to account for model uncertainty.

**Major / general comments:**

Before the article can be published, there is a strong need to clarify its structure. Moreover, the model setup is not well explained or even completely missing (the experiments can not be repeated at all with the given information) and a comprehensive overview of all performed and evaluated simulations is missing. Pieces of information can be gathered from different sections, but this makes it very hard to read.

It would be much more comfortable for the reader to have a section #2 called „Model setup" and a section #3 (or 2.2) called „Performed model simulations" to get a better overview at a first glance.

Following that, the case study should be described in its own Section. More work should be done on that – it is not sufficient just to say that it is a case of „convectively induced precipitation" (beginning of Section 2). The soil-moisture precipitation feedback can depend strongly on the strength of the synoptic-scale forcing. Thus it is essential for the reader to have some idea about the general synoptic conditions of this case.

Related to this: It would be good to include a discussion on the question whether the domain shifting does generate / include new physical processes or not. In Figure 3, for example, LOC 10 00 is shown but not LOC 30 00. In LOC 30 00 and LOC 00 30, the domain is shifted by 30 km, which is not negligible. If a larger part of the ocean / coast is included in the shifted simulation for example, this could very well modify the simulation also in a physical way.

It is strongly recommended to separate the aims and the argumentation for the chosen comparisons from the description of applied methods. The SAL method should be described in a separate section (or subsection) with a clear description of which simulations / precipitation fields (15-min precipitation sums? which evaluation area?) are compared to which reference. Don't mix the argumentation for your method of generating uncertainty estimations into this Section.

In contrast, a discussion of this point is missing in the introduction. Can you give some references / examples of other studies which use domain shiftig to estimate model uncertainty?

Finally, the English language needs to be revised carefully as there are a large number of

inaccuracies. Examples are given below.

**Specific comments**

**1 Introduction**

p. 1, l. 23-24: „Soil moisture affects the partitioning of turbulent heat fluxes …, which once affects the surface temperature"; soil moisture rather directly affects the surface temperature

p. 2, l. 1-2: „in the lower troposphere" → in the boundary layer?

p. 2, l. 2: „surface temperatures can … initiate convection" → can influence the initiation of convection

p. 2., l. 7: „that following the process chain" → that follow the…

p. 2, l. 34-35: „convective precipitation suffers strongly from model uncertainty such (as!) caused by initial and boundary data" → uncertainties  caused by initial and boundary data are not really model uncertainties, even if this is stated in Richard et al. (2007) – is it?

p. 3., l. 1: „many simulations" → a large number of simulations

p. 3, l. 3: „the effect … can be ranged" → can be assessed and quantified?

**2 Soil moisture perturbation and its influence on precipitation**

The Section could be called „Model / experiment setup and overview of performed simulations"

Please give a comprehensive description of the model setup: How many vertical levels did you use? Are the chosen settings for the physics parametrizations similar to the operational ones? For example the parametrization of bare soil evaporation could be decisive for processes in the considered case of convection initiation. Model start time, length of the simulations?

p. 3, Figure 1: It would be great if you could show a larger domain with additional rectangles for used model domains (e.g. black solid line for ctrl domain, black dashed for LOC 00 30 and LOC 30 00).

p. 3, l. 6: „convective introduced precipitation" → convectively induced precipitation

p. 3, l. 8: „A 1 km resolution … provide(s!) a much more accurate simulation of convective precipitation" → more accurate than what?

p. 3, l. 11: „coarse-grid COSMO operational analysis" → 2.8 km is not really coarse; omit „coarse-grid"

p. 3., l. 15: „enhancement [of soil moisture] of 50 %" → did you apply this enhancement taking into

account the underlying soil-type distribution?

p. 3, l. 15: „red framed domain" → insert here „(hereafter, referred to as area „red")"

p. 3, l. 15 ff: „Those changes are **first** applied over whole model domain (DRY_a and MOI_a, Table 1) and second ...**Another** artificial modification is **the** redistribution … (BAND...)" → also give references to names in Table 1 in the following sentences

Figure 2: Is there a reason that you show only 6 out of 8 members of the soil-moisture ensemble?

p. 4, l. 2-3: „high uncertainty of convective precipitation on the initial and boundary data is accounted **for** by..." → **sensitivity** of conv. precip on?

same sentence: better discuss the reasoning behind the method before – either in a separate (sub-)section „Aims and estimation of model uncertainties" or (better) directly at the end of the introduction

p. 4, l. 4: „Those simulation**s**"

p. 4, l. 5: „the simulation with **a domain shifted** by"

p. 4, l. 4 ff (starting with „Here we will focus on..."): This part should be moved into its own (sub-)section (see also general comments); but before, show Table 2 and give the corresponding explanations.
The new (sub)section could be called something like „Overview of convective precipitation event and influence of different soil moisture perturbations".
First, give a more general overview of the case study (Synoptic conditions? When did convection initiation occur? Which processes did contribute? Can you assume in the first place that soil moisture patterns had an influence at all? How much precipitation was observed over which period?).
Only afterwards, sensitivity experiments can be described.

p. 5, l. 1: „differences are predicated to" → presumably caused by? attributed to?

p. 5., l. 1: „brutal changes" → extreme changes?

p. 5., l. 2: „more obvious changes" → modifications / differences?

p. 5, l. 8: „similar order of magnitude **as** soil-moisture modifications"

Table 1, title: „**which** represents the shifting..."

Table 1, last column: The nomenclature „DRY_a *ii jj*" here is not really used in the text; could you give just  „LOC *ii jj*"in this columns and refer to „CTRL-LOC *ii jj*"or „DRY_a-LOC *ii jj*" at places where it is explicitly referenced.

Figure 3, title: „Precipitation rate at 14:45" → this is misleading; I assume that this is the 15-min precipitation sum, recalculated to mm/h (assuming that you have output time steps of 15 min)?

Figure 3: Is there a good reason to use a logarithmic colour scale?
It would be great if you could include the blue rectangle as this evaluation area is used later.

**3 Estimation of model uncertainties**

Section title could be „Determination of objective criteria for the given model uncertainty"

Which precipitation threshold did you use for the SAL (necessary to determine the precipitation objects, called „cells" in this article)?

p. 5, l.12: „provide representative result**s** by using the SAL score" → can you reformulate this sentence?

p. 5, l.13: „for every single time step" → you mean **output time step**? you also have to give it (15 min)?

p. 5, l.13: „The SAL-score **gives**"

p. 7, l. 25 to p. 8, l. 4: as said in the general comments, leave this passage out at this place (parts have to be included when you describe the aims, parts in the Section „Overview of performed model simulations").
The definition of the „uncertainty ensemble" would be clearer if it would be distinguished between the „CTRL-uncertainty ensemble" (shown in Table 2) and the other uncertainty ensembles for the simulations with perturbed soil moisture, e.g. the „DRY_a-uncertainty ensemble".

p. 8., l. 4 ff: Related to the previous comment, it is not easy to understand which simulation is compared to which reference (don't use „CTRL" in l. 8, p. 8 - that ambiguous here; additionally, in the given description of the SAL components, it is called „comp"). How do you count 122 simulations?

Table 1: Columns headings: „lower-left corner" or „LL corner" with abbrev. given in title

Figure 4: Which evaluation area – red or blue? Markers can be hardy distinguished – could you make two sub-plots?

p. 9, l.1-2: „Hence**,** a reduction in **precipitation** amplitude is **related with** too small **and / or** peaked precipitation objects … larger and **/ or** shallow**er** precipitation objects. … This **agrees with**...".

p. 9, l. 6: dependent … „Conclusively, no **systematic behaviour can be detected** for locally perturbed simulations, but for time-shifted simulations, which is caused by the differing precipitation onset."

p. 9., l. 9: „**According** to this definition"

**4 Significant effects of soil moisture modification on precipitation**

p. 9, l. 14 to p. 10, l.2: leave the passage out; as said above, overview of all simulations should be given in Section 2.2 / 3

p. 10, l. 3: „Each uncertainy ensemble will be compared to the CTRL-uncertainty ensemble, only comparing ensemble members with the same domain shifting. That **yields** again a ...“
Again for all output time steps?

p. 10, l. 6: „The percentage of the **values exceeding the uncertainty range** is ...“

p. 10, l. 11: „in only 5 % **of all cases**“

p. 10, l. 14: „soil moisture reduction in the whole domain **(DRY_a)** affects ...“

p. 10., l. 17: „soil moisture enhancement in a sub-domain only **(MOI_p)** ...“

p. 10, l. 19: „the redistribution of soil moisture **as in BAND** does not...“

p. 10, l. 19-20: „The redistribution of soil moisture increases the large-area heterogeneity, but decreases the small-area heterogeneity“ → do you mean that the heterogeneity on the length scale of the chosen band is increased by the perturbation itself while smaller-scale secondary circulations become less important?

Figure 6: Which time steps are analysed? The shading in the rectangles is not necessary and blurs the images. Just give the frames. What are the dashed lines?

**5 Systematics**

p. 12, l. 7: „in MOIST_p, significant but random changes occur“ → are they really random or could the sign of A also be caused by the location of the patch relative to the shifted domain?

p. 13, l. 7: „**According to the z-test** [is it a z-test?], only two simulations [ensembles?] with overall modified soil moisture have a systematic effect...“ → only two of this kind exist; do you mean „only two simulations have a systematic effect: DRY_a and MOI_a, i.e. the two simulations with overall...“

p. 13., l. 11 ff: I would be careful to call it „feedback“ if it is not symmetric. Could the differences of the results found by Barthlott and Kalthoff (2011) compared to the results of others be caused by the influence of orography in their investigation?

---

## Author Comment (AC1) · 4 Aug 2017

We thank the referee for his/her valuable comments and suggestions, which will improve the paper. The responses to your comments are marked in blue.

**General comments**

1. I like the concept of distinguishing between random changes in precipitation and changes that result from soil moisture, but I have some concerns about the representativeness of the results. So far, I am not convinced that the simulation of one day and the evaluation on two comparably small evaluation domains is the right concept. For a more robust conclusion, more cases are necessary and the analysis of one evaluation area alone might be more meaningful. As the authors already mention in the Conclusions, further case studies are needed. If this paper is intended to be a proof-of-concept, the authors should clearly state that in the manuscript and be cautious with any general conclusions.
As far as we know the concept of shifting the model domain has not been conducted in any model study so far. Furthermore, effects from soil moisture changes are rarely compared to other modifications to rate the soil moisture effects. Thus this study aims on proofing the concept of shifting the model domain as an estimate on model uncertainty and on comparing the concept to physically meaningful changes such as modifications in soil moisture. We recommend on enlarging the study to further cases as we could show that the shifted model domain is a useful tool for model uncertainty estimates and that this is necessary to validate the effect of soil moisture. However conducting further case studies is beyond the scope of this work. We will stronger emphases that the major goal was to introduce this concept. Before performing more case studies, one should be aware that the choice of evaluation domain might be crucial for the results and therefore this is a first necessary step before applying the concept to further cases.

2. My main criticism is due to the fact that nothing is being said about the physical processes that are responsible for these differences. The different model runs are compared to each other with the SAL method, but reasons for the differences remain unclear. As the paper is comparably short, I recommend to add a section on the physical processes responsible for the differences. For example, domain averaged time series of convection-related parameters could be shown here.
We included a discussion of several quantities that react systematically on the soil moisture changes in contrast to precipitation.

3. When performing a sensitivity study, the control run has to be evaluated first to assure that it serves as a good basis for the sensitivity runs. I believe you need to insert a subsection on the synoptic controls, the observed precipitation and the results of the control run.
This study aims on the comparison of simulations with different soil moisture and focuses on the differences between various simulations rather than finding the simulation that best fits observational data by model tuning.
We included a section to describe the synoptically conditions and compared it to radar observations. A more detailed evaluation of the model in this case is beyond the scope of this work and would not improve the results of this work.

4. In many operational forecasting centers, soil moisture is already perturbed in their ensemble prediction systems. Some information about that and most importantly, the differences to the method used in this paper, should be added to the manuscript.
Using soil moisture perturbation for ensemble prediction one need to know confidentially that this modification can generate an ensemble spread that is sufficiently large to capture possible realizations. That is exactly what this study addresses by comparing the effect from soil moisture perturbation to perturbation, which does not change physically meaningful parameters.
We included this in the introduction and present some methods of soil moisture modification as conducted by weather services. However, they are not comparable to our soil moisture modification and the comparison to a different uncertainty estimate as they themselves present the uncertainty estimate/model spread.

**Specific comments**

1. What do you mean by model uncertainty? Please clarify. Again, later on: "Only drastic soil moisture changes can exhibit the model uncertainties..." Probably you mean similar uncertainties as in other ensemble systems where e. g. stochastic perturbations are inserted, tuning parameters changed, or different initial and boundary data from another model are used. This has to be made clearer at several locations in the manuscript
   We rewrote the abstract to make this clear

2. P1, L4: "...but the systematic behaviour is still complex..." Up to now, there is no consent about the existence of a systematic relationship of soil moisture to precipitation. I would rather write: "...but the response of precipitation to soil moisture changes is still complex..."
   included

3. P1, L6: Some details about the ensemble approach used in this work should be given here.
   included

4. P1, L23: Surface temperatures are dependent on the sensible heat flux, not the latent heat flux. Please rephrase.

5. You mean the water content of air? Then it's probably better to write: "Secondly, soil moisture strongly influences the low-level humidity via the latent heat flux."
   included

6. "...react on the soil moisture." Better: "...depend on soil moisture due to its effects on low-level temperature and humidity."
   included

7. What do you mean with the synergy of soil moisture-precipitation feedbacks?
   The role of orography in the soil-moisture feedback

8. Is shallow convection still parameterized? Which COSMO version do you use?
   Yes, Version 4.4

9. P3: Is the total drying of the soil the respective permanent wilting point?
   With the 50% increase in soil moisture, did you assure that you don't have larger values than the porosity allows?
   Values of soil moisture are set to zero in the initial conditions and multiplied by a factor of 1.5 independent of wilting point and porosity, respectively
   You state in the manuscript that you want to show the full range of soil moisture influence. So why did you use just a 50% increase and not the maximum value possible for the respective soil type?
   It is right that an enhancement of 50% does not show the full range of possible soil moisture influence. We corrected this statement. Anyhow, the increase by 50% exceeds the increased that is reached by using the relatively wet soil moisture pattern from another day so that this state a large change as the realistic modifications. We included a figure to show this.
   Did you change all levels in the soil in the same way?
   Yes
   Did you make the changes at the model initialization time?
   Yes, we added this in the text.

10. P4, Figure 2: This figure is too pixelated, the text is hardly readable.
    Concerning your band pattern: Does the soil moisture changes from 1 grid point to the other or is there a smoother transition over a couple of grid points? Do these strong gradients introduce any thermal direct circulations?
    We do not have a smooth transition zone, but the initial soil moisture conditions also include strong gradients.

11. Which moist simulation do you refer to? I don't agree with the statement that in the moist simulation, precipitation occurs mainly at places that are free of precipitation in the CNTRL run. At least, I don't see that in Figure 3.
    We included a more detailed description of where precipitation occurs at the shown timestep

12. Figure 3: Instead of showing one time of day, a 24-h accumulated precipitation would be much more meaningful. Soil moisture may also influence the timing of cloud formation, so

one snapshot might not be enough to show the overall effect. In addition, time series of domain-averaged precipitation should be shown as well

We added the timeseries of accumulated precipitation.

13. Random perturbations are introduced by shifting the domain boundaries. Please explain in more detail, why you consider this as random perturbations. One way to prove that would be to insert stochastic perturbations e. g. in the initial temperature field. The authors should comment on that.

With this method we can ensure, that we do not generate any other patterns, that are overlayed and do not change meaningful quantities such as temperature. We described this more detailed.

**Technical comments**

We corrected the manuscript as suggested in the technical comments

---

## Author Comment (AC2) · 4 Aug 2017

We thank the referee for his/her valuable comments and suggestions, which will improve the paper. The responses to your comments are marked in blue.

Review on „Assessing the uncertaintiy of soil-moisture impact on convective precipitation by an ensemble approach" by O. Henneberg, F. Ament, and V. Grützun

In this article, the authors evaluate the impact of different soil-moisture initializations on the simulation of convective precipitation with the COSMO model, using a set of ensemble simulations for one case study. These consist of 8 uncertainty ensembles based on one soil-moisture ensemble.

The uncertainty ensembles consist of 11 simulations with a shifted model domain and in one case, on 6 additional simulations with a modified start time.

The underlying idea is that the uncertainty ensemble is a method to estimate model uncertainty, which is then used to assess the significance of different soil-moisture initializations. It is found that „only drastic soil moisture changes" can overcome the model uncertainties.

The idea to compare model uncertainty with soil-moisture induced uncertainty seems to be somehow neglected in recent literature and therefore the article is recommended for publication, even if it is not entirely clear if domain shifting can be regarded as a reliable measure to account for model uncertainty.

**Major / general comments:**

Before the article can be published, there is a strong need to clarify its structure. Moreover, the model setup is not well explained or even completely missing (the experiments can not be repeated at all with the given information) and a comprehensive overview of all performed and evaluated simulations is missing. Pieces of information can be gathered from different sections, but this makes it very hard to read.

It would be much more comfortable for the reader to have a section #2 called „Model setup" and a section #3 (or 2.2) called „Performed model simulations" to get a better overview at a first glance.

Following that, the case study should be described in its own Section. More work should be done on that – it is not sufficient just to say that it is a case of „convectively induced precipitation" (beginning of Section 2). The soil-moisture precipitation feedback can depend strongly on the strength of the synoptic-scale forcing. Thus it is essential for the reader to have some idea about the general synoptic conditions of this case.

We restructured the article as following:
  2 Modelling approach
    2.1 Numerical Setup
    2.2 Soil moisture experiments
    2.3 Ensemble approach
  3 Case description
  4 Results
    4.1 Estimate of model uncertainty ...
and extended the description of the synoptic situation

Related to this: It would be good to include a discussion on the question whether the domain shifting does generate / include new physical processes or not. In Figure 3, for example, LOC 10 00 is shown but not LOC 30 00. In LOC 30 00 and LOC 00 30, the domain is shifted by 30 km, which is not negligible. If a larger part of the ocean / coast is included in the shifted simulation for example, this could very well modify the simulation also in a physical way.

We agree, that a larger fraction of sea surface in the model domain includes physical processes due to different surface fluxes that affect precipitation formation. However we do not see any trend that a stronger shifting changes precipitation in one or the other direction and therefore consider the shifting to be suitable to generate changes. We included a discussion paragraph on that.
It is strongly recommended to separate the aims and the argumentation for the chosen comparisons from the description of applied methods. The SAL method should be described in a separate section (or subsection) with a clear description of which simulations / precipitation fields (15-min precipitation sums? which evaluation area?) are compared to which reference. Don't mix the argumentation for your method of generating uncertainty estimations into this Section.

We tried to make clearer that SAL is applied on every model output step of the precipitation rate. However, we tested the analysis for the 15min accumulated precipitation for the REAL ensembles and found the results are more sensitive to the chosen analysis area than the chosen output variable.

In contrast, a discussion of this point is missing in the introduction. Can you give some references / examples of other studies which use domain shifting to estimate model uncertainty?

We are not aware of any studies using this approach

Finally, the English language needs to be revised carefully as there are a large number of inaccuracies. Examples are given below.

We included the following comments either as suggested or as stated in blue:

**Specific comments**

**1 Introduction**

p. 1, l. 23-24: „Soil moisture affects the partitioning of turbulent heat fluxes …, which once affects
        included
the surface temperature"; soil moisture rather directly affects the surface temperature
p. 2, l. 1-2: „in the lower troposphere" → in the boundary layer?
        included
p. 2, l. 2: „surface temperatures can … initiate convection" → can influence the initiation of convection
        included
p. 2., l. 7: „that following the process chain" → that follow the…
        included
p. 2, l. 34-35: „convective precipitation suffers strongly from model uncertainty such (as!) caused by initial and boundary data" → uncertainties caused by initial and boundary data are not really model uncertainties, even if this is stated in Richard et al. (2007) – is it?
        Reformulated
p. 3., l. 1: „many simulations" → a large number of simulations
        included
p. 3, l. 3: „the effect … can be ranged" → can be assessed and quantified?
        Included

**2 Soil moisture perturbation and its influence on precipitation**

The Section could be called „Model / experiment setup and overview of performed simulations"
Please give a comprehensive description of the model setup: How many vertical levels did

you use?

Are the chosen settings for the physics parametrizations similar to the operational ones? For example the parametrization of bare soil evaporation could be decisive for processes in the considered case of convection initiation. Model start time, length of the simulations?

p. 3, Figure 1: It would be great if you could show a larger domain with additional rectangles for used model domains (e.g. black solid line for ctrl domain, black dashed for LOC 00 30 and LOC 30 00).

> We extenden the figure for simulation CTRL to a slighly larger domain, to show both analysis areas within this plot

p. 3, l. 6: „convective introduced precipitation" → convectively induced precipitation

> included

p. 3, l. 8: „A 1 km resolution … provide(s!) a much more accurate simulation of convective precipitation" → more accurate than what?

> Than simulations for which a convection pramatrization is required

p. 3, l. 11: „coarse-grid COSMO operational analysis" → 2.8 km is not really coarse; omit „coarsegrid"

> included

p. 3., l. 15: „enhancement [of soil moisture] of 50 %" → did you apply this enhancement taking into account the underlying soil-type distribution?

> No. Are there essential reasons to do so?

p. 3, l. 15: „red framed domain" → insert here „(hereafter, referred to as area „red")"

> included

p. 3, l. 15 ff: „Those changes are first applied over whole model domain (DRY_a and MOI_a, Table 1) and second ...Another artificial modification is the redistribution … (BAND...)" → also give references to names in Table 1 in the following sentences

> included

Figure 2: Is there a reason that you show only 6 out of 8 members of the soil-moisture ensemble?

> We included a rectangle for the locally changed soil moisture. That differs from the allover changes by its local restriction.

p. 4, l. 2-3: „high uncertainty of convective precipitation on the initial and boundary data is accounted for by..." → sensitivity of conv. precip on?

> reformulated

same sentence: better discuss the reasoning behind the method before – either in a separate (sub-)section „Aims and estimation of model uncertainties" or (better) directly at the end of the introduction

> We included in the introduction that we used this estimate on model uncertainty to compare this to comparable strong soil moisture changes as soil moisture perturbations are also used in ensemble prediction.

p. 4, l. 4: „Those simulations"

> included

p. 4, l. 5: „the simulation with a domain shifted by"

> included

p. 4, l. 4 ff (starting with „Here we will focus on..."): This part should be moved into its own (sub-)section (see also general comments); but before, show Table 2 and give the corresponding explanations.

The new (sub)section could be called something like „Overview of convective precipitation event and influence of different soil moisture perturbations".

First, give a more general overview of the case study (Synoptic conditions? When did convection initiation occur? Which processes did contribute? Can you assume in the first place that soil moisture patterns had an influence at all? How much precipitation was

observed over which period?).

Only afterwards, sensitivity experiments can be described.

We included a section on the synoptic description and show the occurrence of precipitation in the radar

p. 5, l. 1: „differences are predicated to" → presumably caused by? attributed to?

included

p. 5., l. 1: „brutal changes" → extreme changes?

included

p. 5., l. 2: „more obvious changes" → modifications / differences?

included

p. 5, l. 8: „similar order of magnitude as soil-moisture modifications"

included

Table 1, title: „which represents the shifting..."

included

Table 1, last column: The nomenclature „DRY_a ii jj" here is not really used in the text; could you give just „LOC ii jj"in this columns and refer to „CTRL-LOC ii jj"or „DRY_a-LOC ii jj" at places where it is explicitly referenced.

ii and jj is a replacement for all simulations in table to as described in the heading.

Figure 3, title: „Precipitation rate at 14:45" → this is misleading; I assume that this is the 15-min precipitation sum, recalculated to mm/h (assuming that you have output time steps of 15 min)?

It's the precipitation rate and results are very different chosing one or the other variable

Figure 3: Is there a good reason to use a logarithmic colour scale?

We decided to use a logarythmic scale to show precipitation detailes, which cant be seen with a linear scale (which is anyhow rather unusal for precipitation). And a logarythmic scale is still more intuitive than a irregular scale.

It would be great if you could include the blue rectangle as this evaluation area is used later.

Included

**3 Estimation of model uncertainties**

Section title could be „Determination of objective criteria for the given model uncertainty"
Which precipitation threshold did you use for the SAL (necessary to determine the precipitation objects, called „cells" in this article)?

The treshold for every object is calculated by the 5%-percentile of all precipitating grid points with rates higher than 1e-4 kg m-2 s-1

p. 5, l.12: „provide representative results by using the SAL score" → can you reformulate this sentence?

Changed

p. 5, l.13: „for every single time step" → you mean output time step? you also have to give it (15 min)?

changed

p. 5, l.13: „The SAL-score gives"

provides

p. 7, l. 25 to p. 8, l. 4: as said in the general comments, leave this passage out at this place (parts have to be included when you describe the aims, parts in the Section „Overview of performed model simulations").

The definition of the „uncertainty ensemble" would be clearer if it would be distinguished between the „CTRL-uncertainty ensemble" (shown in Table 2) and the other uncertainty ensembles for the simulations with perturbed soil moisture, e.g. the „DRY_a-uncertainty

ensemble".

> We renamed into reference simulation for ensemble and ensemble generating changes.

p. 8., l. 4 ff: Related to the previous comment, it is not easy to understand which simulation is compared to which reference (don't use „CTRL" in l. 8, p. 8 - that ambiguous here; additionally, in the given description of the SAL components, it is called „comp"). How do you count 122 simulations?

> We changed the subscrips in SAL to ctrl and diff. There are 122 combinations to compare the different simulations. We corrected that.

Table 1: Columns headings: „lower-left corner" or „LL corner" with abbrev. given in title

> included

Figure 4: Which evaluation area – red or blue?

> Red, we included it in the text now

Markers can be hardly distinguished – could you make two sub-plots?

> included

p. 9, l.1-2: „Hence, a reduction in precipitation amplitude is related with too small and / or peaked precipitation objects … larger and / or shallower precipitation objects. … This agrees with...".

> included

p. 9, l. 6: dependent … „Conclusively, no systematic behaviour can be detected for locally perturbed simulations, but for time-shifted simulations, which is caused by the differing precipitation onset."

> included

p. 9., l. 9: „According to this definition"

> included

**4 Significant effects of soil moisture modification on precipitation**

p. 9, l. 14 to p. 10, l.2: leave the passage out; as said above, overview of all simulations should be given in Section 2.2 / 3

> deleted

p. 10, l. 3: „Each uncertainy ensemble will be compared to the CTRL-uncertainty ensemble, only comparing ensemble members with the same domain shifting. That yields again a ...“
Again for all output time steps?

> Yes included

p. 10, l. 6: „The percentage of the values exceeding the uncertainty range is ...“
p. 10, l. 11: „in only 5 % of all cases"
p. 10, l. 14: „soil moisture reduction in the whole domain (DRY_a) affects ...“
p. 10., l. 17: „soil moisture enhancement in a sub-domain only (MOI_p) ...“
p. 10, l. 19: „the redistribution of soil moisture as in BAND does not...“

> all included

p. 10, l. 19-20: „The redistribution of soil moisture increases the large-area heterogeneity, but decreases the small-area heterogeneity" → do you mean that the heterogeneity on the length scale of the chosen band is increased by the perturbation itself while smaller-scale secondary circulations become less important?

> Changed to: The redistribution of soil moisture changes the heterogeneity of the soil moisture by reducing small-scale structures, but induce stronger variations on the large scale.

Figure 6: Which time steps are analysed? The shading in the rectangles is not necessary and blurs the images. Just give the frames. What are the dashed lines?

> Shading was deleted, the dashes lines represent average values

**5 Systematics**

p. 12, l. 7: „in MOIST_p, significant but random changes occur" → are they really random or could the sign of A also be caused by the location of the patch relative to the shifted domain?

> You mean accorind to the location of the modification (patch) relative to the domain boundaries? The location of the patch stays the same.

p. 13, l. 7: „According to the z-test [is it a z-test?], only two simulations [ensembles?] with overall modified soil moisture have a systematic effect..." → only two of this kind exist; do you mean „only two simulations have a systematic effect: DRY_a and MOI_a, i.e. the two simulations with overall..."

> This had been formulated missleading. We reformulated this.

p. 13., l. 11 ff: I would be careful to call it „feedback" if it is not symmetric. Could the differences of the results found by Barthlott and Kalthoff (2011) compared to the results of others be caused by the influence of orography in their investigation?

> What is menat by symmetric? There are mainly two feedback mechanism, one positive one negative resulting dependent on the conditions in either an overall positive or negative feedback. As in all reffered manuscripts this is called feedback we will stick to the term as well. The results from Barthlott are influenced from the orography.

---

## Author Response (AR2)

**Answer to reviewer 1:**

We thank the referee for the suggested correction and included them or rephrased the sentences:

- I suggest to change the title: "impacts" instead of "impact" because soil moisture does not have just one possible impact on the atmosphere - included
- P1, L15: "Increased" instead of "Enhanced" - included
- P1, L16: "predominantly" - included
- P2, L1: "stays" instead of "states" - included
- P2, L6: You cannot say that "soil moisture SITS at the beginning...", please rephrase. - Rephrased
- P2, L16-18: "Whereas, ..." Please check the grammar of that sentence, it does not make sense to me. - rephrased
- P2, L28: "synoptically" - changed
- P7, Table 1 caption: "gridpoints" should be "grid points" - included
- P9, L7: "eg" should be "e.g."; included "circulation" should be "circulations" - included
- P10, L3: "case study", not "case-study" - included
- P10, Figure 7: What is the red dot in the upper panel around 21 UTC? - Deleted red dot.

[revised manuscript text omitted]